# The association between depression and later educational attainment in children and adolescents: a systematic review protocol

Alice Wickersham [1,2] Sophie Epstein [1,2] Holly Victoria Rose Sugg [3] Robert Stewart [1,2] Tamsin Ford [4] Johnny Downs [1,2]

[1]Institute of Psychiatry, Psychology and Neuroscience, King's College London, London, UK
[2]NIHR Maudsley Biomedical Research Centre, South London and Maudsley NHS Foundation Trust, London, UK
[3]University of Exeter Medical School, University of Exeter, Exeter, UK
[4]Department of Psychiatry, University of Cambridge, Cambridge, UK

**Correspondence to**
Alice Wickersham;
alice.wickersham@kcl.ac.uk

## ABSTRACT

**Introduction** Depression represents a major public health concern for children and adolescents, and is thought to negatively impact subsequent educational attainment. However, the extent to which depression and educational attainment are directly associated, and whether other factors play a role, is uncertain. Therefore, we aim to systematically review the literature to provide an up-to-date estimate on the strength of this association, and to summarise potential mediators and moderators on the pathway between the two.

**Methods and analysis** To identify relevant studies, we will systematically search Embase, PsycINFO, PubMed, Education Resources Information Centre and British Education Index, manually search reference lists and contact experts in the field. Studies will be included if they investigate and report on the association between major depression diagnosis or depressive symptoms in children and adolescents aged 4–18 years (exposure) and later educational attainment (outcome). Two independent reviewers will screen titles, abstracts and full texts according to eligibility criteria, perform data extraction and assess study quality according to a modified version of the Newcastle-Ottawa Scale. If sufficiently homogeneous studies are identified, summary effect estimates will be pooled in meta-analysis, with further tests for study heterogeneity, publication bias and the effects of moderators using meta-regression.

**Ethics and dissemination** Because this review will make use of already published data, ethical approval will not be sought. The review will be submitted for publication in a peer-reviewed journal, presented at practitioner-facing conferences, and a lay summary will be written for non-scientific audiences such as parents, young people and teachers. The work will inform upcoming investigations on the association between child and adolescent mental health and educational attainment.

**PROSPERO registration number** CRD42019123068

## Strengths and limitations of this study

► This review will provide a timely update on the association between depression and later educational attainment, and will be the first to summarise evidence on both moderators and mediators of the association.
► A comprehensive search strategy is planned, including searches of both health and education electronic databases, forward and backward citation searching, and contacting authors and experts in the field.
► The exclusion of grey literature and studies not published in the English language may cause some relevant studies to be missed.
► However, limiting the review to studies which make use of standardised depression measures and academic records will ensure that included studies are of a reasonable quality, and reduce heterogeneity for meta-analysis.
► Including prospective longitudinal studies may also aid inference of a causal direction between depression and later educational attainment.

## INTRODUCTION

Depression is a mental health disorder which, in children and young people, is particularly characterised by symptoms such as low mood, irritability, negative self-perceptions, reduced energy, sleep disturbances and cognitive problems.[1 2] It is a major public health concern for this age group, with a recent study estimating that 2.1% of children and adolescents in England meet criteria for a depressive disorder.[3] Depression is thought to predict a range of negative psychosocial outcomes, including poorer school outcomes such as educational attainment. A meta-analysis of longitudinal studies by Riglin *et al*[4] suggests that depression, and to some extent anxiety, are associated with attainment outcomes, including the failure to complete compulsory education and low school grades. This may in turn have long-term and far reaching negative consequences, with poor school performance predicting unemployment, homelessness, poor health and suicide attempt.[5–8]

In spite of this, educational systems have historically shown reluctance to divert

resources from traditional teaching towards mental health provision.[9][10] But school-based mental health provision is growing, and the need for a thorough and robust understanding of how depression can impact educational attainment is of critical importance to guide this growing area.[11][12] In particular, a range of candidate factors are thought to affect school performance and may mediate or moderate the impact of depression on educational attainment. Knowledge of these could be used to tailor mental health provision and highlight priority groups for intervention. Candidate factors might include executive function, sleep, classroom environment and engagement.[13–15] Parents and families are also likely to play a critical role in the association, with socioeconomic status, the family environment, parent involvement, parental education and parental mental health all thought to be associated with school outcomes.[16–19]

At the time of Riglin *et al*'s review,[4] comparatively few studies had been carried out outside of North America. Additionally, while the presence of depressive symptoms remains an exposure of interest, Riglin *et al* were unable to draw comparisons of studies to examine the effect of clinical depression on grades, due to an insufficient number of such studies being available. These gaps in the literature may have since been addressed. Therefore, the first aim of our systematic review is to revisit this association and provide an up-to-date estimate on the strength of the relationship between child and adolescent depression and later educational attainment.

Beyond an examination of the moderating effects of age, gender, country and length of study follow-up period, Riglin *et al* considered evidence on the pathways between depression and educational attainment to be beyond the scope of their systematic review. Therefore, the second aim of our systematic review is to investigate mediators and moderators in the relationship between child and adolescent depression and later educational attainment, with the intention of proposing a pathway model between the two.

## METHODS AND ANALYSIS

This protocol follows the Preferred Reporting Items for Systematic Reviews and Meta-Analyses (PRISMA) reporting guidelines for systematic review and meta-analysis protocols (online supplementary file 1).[20] The final review will also follow PRISMA reporting guidelines, and will include a PRISMA checklist and flow diagram.[21]

### Eligibility criteria

#### Population

Inclusion and exclusion criteria are summarised in table 1. Because of the relationship under study, we are primarily interested in depression during the school years. Therefore, participants in included studies will be children and adolescents, all within the 4–18 year age range at the time of exposure measurement, encompassing the compulsory school age range in most countries. Studies will make use of data from countries with compulsory educational policies (as determined from the countries' government or public sector websites).[22] In countries without such policies, school attendance is likely to be poorer for children with mental health difficulties such as depression, and therefore, attainment data are less likely to be adequately available for this group. Studies will be excluded if they recruit participants from post-secondary education settings such as universities at baseline, to ensure that results pertain to depression in childhood and adolescence during school. No further

| Table 1 | PECOS criteria for inclusion and exclusion of studies | |
|---|---|---|
| **Parameter** | **Inclusion criteria** | **Exclusion criteria** |
| Population | ► Participants aged 4–18 years (inclusive) at the time of exposure measurement.<br>► Countries with compulsory education policies. | ► Participants recruited from postsecondary education settings. |
| Exposure | ► Depression diagnosis or depressive symptoms as measured using a standardised diagnostic measure or a named measurement instrument. | ► Internalising symptoms or other affective disorders such as bipolar disorder. |
| Comparison | No restrictions | No restrictions |
| Outcome | ► Educational attainment as measured using academic or administrative records. | ► School drop-out, general intelligence, aptitude, or ability. |
| Study design | ► Investigate and report results on the relationship between depression diagnosis or depressive symptoms (exposure) and later educational attainment (outcome).<br>► Quantitative longitudinal studies with prospective data collection.<br>► Original research published in a peer-reviewed journal.<br>► Published in English.<br>► Full text available including data on the association between depression and educational attainment. | ► Aim to conduct or evaluate an intervention during the observed study period. |

restrictions will be placed on study setting. Age at the time of outcome measurement will not be restricted, such that attainment at higher education completed during adulthood will be included.

## Exposure and outcome variables

Included studies will investigate and report results on the relationship between child or adolescent depression (exposure) and later educational attainment (outcome). Child or adolescent depression will be operationalised as depressive symptoms or depression diagnosis as identified using a standardised diagnostic measure or a named measurement instrument. Studies of internalising symptoms or other depressive disorders, such as bipolar disorder, will not be included. We anticipate that studies will use various measures of educational attainment, including but not limited to highest level of education completed, and standardised academic assessment scores. Studies will not be included if they do not measure educational attainment using academic or administrative records, instead relying on an unclear source or on self-reported, parent-reported or teacher only-reported outcomes. The accuracy of informant-rated academic performance can be mixed compared with results from externally validated assessments.[23] School drop-out will not be considered in this review as it represents a distinct construct which may have different mechanisms associated with it; indeed poor educational attainment is thought to be one of the predictors of school drop-out.[24] Measures of general intelligence, aptitude or ability will also not be included.

## Study design

Only quantitative, prospective longitudinal studies will be included. This can aid the inference of a causal direction, particularly where included studies adjust for covariates such as prior attainment.[25 26] Reviews, meta-analyses, cross-sectional studies, retrospective studies, case reports, clinical vignettes, randomised controlled trials and exclusively qualitative studies will, therefore, be excluded. Studies which aim to conduct or evaluate an intervention during the observed study period will also be excluded to ensure that any observed association is not influenced by the intervention. Studies will be included if they are original research (therefore, excluding editorials, opinion pieces, letters to the editor and commentaries), are published in English and are published in a peer-reviewed journal (therefore, excluding grey literature, books, chapters, theses, dissertations and conference proceedings). The full text must also be available (including data on the association between depression and educational attainment). Corresponding authors will be contacted for full texts that cannot be obtained publicly or via King's College London's institutional access (two email attempts will be made). To maximise the number of relevant studies captured, no date range will be imposed on the search.

## Information sources

Studies will be identified by searching the following electronic databases:
- ► Embase (via Ovid).
- ► PsycINFO (via Ovid).
- ► PubMed (via NCBI).
- ► Education Resources Information Centre (via EBESCO).
- ► British Education Index (via EBESCO).

We will also search reference lists of included studies and relevant existing reviews (backward citation searching), and papers which have referenced them (forward citation searching). Backward and forward citation searching will be carried out in Web of Science Core Collection. If a citation cannot be identified in Web of Science, Google Scholar will be searched. Finally, experts in the field and corresponding authors of included studies will be contacted with a link to the PROSPERO record detailing eligibility criteria to identify any additional papers (two email attempts will be made).

## Search strategy for electronic databases

Electronic database searching will be conducted using a combination of key words (using truncation as needed) and subject headings (exploded to include narrower terms). The exact search terms used will, therefore, be adapted according to database thesauruses, but broadly will be grouped according to three concepts:
1. Age (eg, child, adolescent, youth).
2. Educational attainment (eg, academic performance, educational attainment, school failure).
3. Depression (eg, depression, depressive disorders)

English language limits will be applied. The full search strategy for Embase can be found in table 2.

| # | Search terms |
|---|---|
| Table 2 | Full search strategy for Embase |
| 1 | exp adolescent/ OR exp adolescence/ OR exp child/ OR exp childhood/ OR child*.tw OR adolescen*.tw OR teenag*.tw OR youth*.tw OR (young adj (people or person)).tw |
| 2 | Limit 1 to english language |
| 3 | exp academic achievement/ OR exp outcome of education/ OR ((academic or educational or school or classroom) adj (achievement or performance or attainment or success or failure)).tw |
| 4 | Limit 3 to english language |
| 5 | exp depression/ OR depressi*.tw |
| 6 | Limit 5 to english language |
| 7 | #2 AND #4 AND #6 |

Exp (search term)/denotes exploding a subject heading; .tw denotes searching for a key word in the title, abstract and drug trade name.
*Denotes truncation.

## Data management

All identified citations will be downloaded and managed in EndNote, and duplicates will be removed. Article screening and data extraction will be tracked using Microsoft Excel.

## Selection process

Following electronic database searching, initial title and abstract screening will be carried out by two independent reviewers. The reviewers will initially screen 10% of the titles and abstracts and agreement will be checked, before proceeding to screen all the remaining titles and abstracts according to the prespecified eligibility criteria (table 1). All references will be screened by both reviewers to ascertain the level of agreement. Articles which appear eligible from the abstract, or are of unclear eligibility, will pass to full-text screening. This will also be carried out by two independent reviewers, again following an initial 10% screen to check agreement. The process of independent abstract and full-text screening will be repeated for references identified during backward and forward citation searching following an initial screen carried out by the lead researcher. Any disagreements over article eligibility will be discussed, and a third reviewer will be consulted if a consensus cannot be reached.

## Data extraction

Data will be extracted using a data extraction form which will be informed by the full-text screening and will be piloted on the included studies before being finalised. The anticipated data extraction form is in online supplementary file 2. Data extraction will be carried out by two reviewers. Any disagreements over data extraction will be discussed, and a third reviewer will be consulted if a consensus cannot be reached. If multiple studies use the same data sources, they will still be recorded separately in data extraction as they may offer insights to different covariates, mediators and moderators.

## Quality assessment

The included studies will be assessed for risk of bias at the study level using a modified version of the Newcastle-Ottawa Scale (NOS) for cohort studies.[27] The NOS is recommended by the Cochrane Handbook (section 13.5.2.3).[28] Some items will be adapted for relevance to this review (eg, making reference to depression, educational attainment and schools), and two items on sample size and statistical tests will be added similar to a previous systematic review on education and mental health.[29] The comparability of groups with and without depression will primarily be assessed based on whether the study controls for both age and gender, either in the selection of the cohort or in adjusted or stratified analyses. An additional point will be given to studies which control for any other factor.

Risk of bias will be assessed by two independent reviewers. Risk of bias assessment will be conducted on 10% of the included studies and agreement checked before proceeding to the remaining studies. Disagreements will be discussed, and a third reviewer will be consulted if a consensus cannot be reached. Results from the risk of bias assessment will be taken into consideration when interpreting the strength of evidence for the reported associations, and will be considered in meta-analysis (see the 'Data Synthesis' section).

## Data synthesis

The characteristics and findings of included studies will be presented in a data extraction table and will be discussed in a narrative synthesis. If multiple studies are identified which investigate similar exposure and outcome variables, a random-effects meta-analysis will be conducted. If multiple articles measure the same association in an identical cohort, and if multiple relevant associations are reported within one article, the mean of these associations will be taken and used in meta-analysis.

Summary estimates for the effect of depression on later educational attainment will be pooled. Estimates from multivariable analyses which control for at least one covariate will be used, unless insufficient studies conduct comparable multivariable analyses, in which case estimates from bivariate analyses will be used instead. A CI and p value will be calculated for the pooled effect estimate. Heterogeneity of effect estimates will be investigated using the $I^2$ statistic. Publication bias will be assessed using funnel plots and Egger's test for publication bias if at least ten studies are included in the meta-analysis (Cochrane Handbook section 10.4.3.1).[28]

To investigate moderators in the association between depression and educational attainment, subgroup analyses will be carried out using meta-regression. As with Riglin *et al*,[4] we will examine the moderating effects of age, gender and follow-up period. In addition, the impact of risk of bias will be investigated. Other investigations of potentially important moderators may be informed post hoc by the included studies, but will be identified as such in the final report as recommended in the Cochrane Handbook (section 9.6.5.2).[28] Candidate moderators include ethnicity, socioeconomic status, variables relating to the parent or family context, comorbidities, country, the measurement of clinical diagnosis versus symptomatology at exposure, and whether studies are conducted in clinical versus community settings. As with the main meta-analysis, this will only be possible where similar moderator variables or analyses are available for multiple studies (eg, if participant characteristics are measured on similar scales and adjusted for at similar time points). Finally, if multiple studies investigate similar mediator variables, meta-analytical structural equation modelling will be employed to synthesise their effects.

## Study status

Initial electronic database searching was conducted in November 2018. The search will be updated prior to completion, with the review expected to be completed in December 2019.

## ETHICS AND DISSEMINATION

On completion, the review will be submitted to a peer-reviewed journal in the field of mental health or educational research for publication. Findings will also be presented at practitioner-facing conferences, and a lay summary will be written for non-scientific audiences such as parents, young people and teachers. The findings will inform upcoming work on the association between child and adolescent mental health and educational attainment.

### Patient and public involvement

This review has been planned to support ongoing health and education data linkage work which has been carried out in consultation with several patient and caregiver groups at the National Institute for Health Research Biomedical Research Centre, South London and Maudsley National Health Service Foundation Trust and King's College London.[30 31] The results of the review and other research will be discussed with a young person's mental health advisory group, parents and teachers to guide upcoming work with linked health and education data.

## DISCUSSION

While our methodology is informed by Riglin et al,[4] we have made several modifications to this design which we believe will strengthen the review. First, we will broaden our information sources to include Embase and British Education Index. Second, we will examine the 4–18 year age range rather than the 8–18 year age range, as this captures the compulsory school years in most countries. Third, we will not limit our review to community-based samples. While findings from clinical samples are of limited generalisability, this will maximise the amount of available evidence that is captured on the association, particularly with regard to diagnosed depressive disorders.

In our review, we will focus on depression and will not report data on anxiety or other internalising disorders. Riglin et al noted that the association between anxiety and educational attainment is much less clear, with the possibility that anxiety may sometimes have a positive role. Therefore, the pathways leading to educational attainment outcomes are likely to be different for anxiety and depression, and should be considered separately. Finally, to ensure the objectivity of our outcome measure, we will exclude studies which do not obtain educational attainment from academic or administrative records.

This review inevitably will have some limitations. Our focus is on attainment subsequent to depression, such that research on the more nuanced relationships between depression and education over time are beyond the scope of this study. Our restriction to longitudinal studies may also result in some relevant data being missed (for instance from case–control studies or the control groups of randomised controlled trials), however, this is considered an important inclusion criteria as longitudinal designs are well suited for investigating the relationships under study and can aid causal inference.[25] The exclusion of studies not published in the English language may omit some studies of this association, however, we lack resources for translation. Additionally, the exclusion of grey literature may also cause relevant findings to be missed.

This systematic review will provide a synthesis of the available evidence on the association between child and adolescent depression and later educational attainment. This work is timely and of great public interest. Recent studies demonstrate the high prevalence of mental health problems in the child and adolescent age group[3] and ongoing plans to improve child mental health provision in schools demonstrate the increasing recognition that education and well-being are closely linked.[11] Understanding whether and how depression influences later educational attainment is critical to developing effective interventions for affected groups. To our knowledge, this will be the first systematic review to provide a focused synthesis on child and adolescent depression and academic records of educational attainment, with the additional aim of investigating mediators and moderators in order to propose a comprehensive pathway model on the association between the two.

**Acknowledgements** With thanks to Simon Benham-Clarke and Darren Moore for providing comments on the search strategy, and to Aida Bonham for assisting during the early stages of the review.

**Contributors** AW (guarantor) coconceived and designed the review, developed the search strategy and wrote the initial manuscript. SE advised on study design and provided revisions on the manuscript. HVRS provided revisions on the manuscript. RS, TF and JD coconceived the review, advised on study design and provided revisions on the manuscript.

**Funding** This paper represents independent research funded by the National Institute for Health Research (NIHR) Biomedical Research Centre at South London and Maudsley NHS Foundation Trust and King's College London. SE has been employed as an NIHR Academic Clinical Fellow and has received salary support from an MQ Data Science Award and from the Psychiatry Research Trust. HVRS is funded by the University of Exeter Medical School. JD is supported by NIHR Clinician Science Fellowship award (CS-2018-18-ST2-014) and has received support from a Medical Research Council (MRC) Clinical Research Training Fellowship (MR/L017105/1) and Psychiatry Research Trust Peggy Pollak Research Fellowship in Developmental Psychiatry.

**Disclaimer** The views expressed are those of the authors and not necessarily those of the NHS, the NIHR or the Department of Health and Social Care.

**Competing interests** None declared.

**Patient consent for publication** Not required.

**Ethics approval** This review will make use of already published data, therefore, ethical approval will not be sought.

**Provenance and peer review** Not commissioned; externally peer reviewed.

**ORCID iDs**
Alice Wickersham http://orcid.org/0000-0002-7402-7690
Sophie Epstein http://orcid.org/0000-0002-2118-908X
Holly Victoria Rose Sugg http://orcid.org/0000-0001-8530-2726
Robert Stewart http://orcid.org/0000-0002-4435-6397
Tamsin Ford http://orcid.org/0000-0001-5295-4904

Johnny Downs http://orcid.org/0000-0002-8061-295X

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
