## [Reviewer comments · BMJ Open]

ARTICLE DETAILS

TITLE (PROVISIONAL)	The association between depression and later educational attainment in children and adolescents: A systematic review protocol
AUTHORS	Wickersham, Alice; Epstein, Sophie; Sugg, Holly; Stewart, Robert; Ford, Tamsin; Downs, Johnny

VERSION 1 – REVIEW

REVIEWER	Mary N Haan UCSF School of Medicine, San Francisco, CA USA
REVIEW RETURNED	01-Jun-2019

GENERAL COMMENTS	This is a plan for a systematic review of the literature about the association between childhood depression and later educational attainment. The review is expected to be completed by July 2019. Selection criteria for studies are appropriate and result in the inclusion of only longitudinal studies. Data on parents is not included in the description; this could be essential in that the childhood environment is known to powerfully affect subsequent mental health and other outcomes. The parental social context, parental history of depression/depressive symptoms, early parental death/abandonment/rejection, and childhood socioeconomic status are all of critical importance. These are not mentioned in the review. The influence of educational attainment is intended to be an outcome in this study; however, education commences at age 4-5 and mental health context should be modeled as a time dependent variable.
--

REVIEWER	Lorraine McKelvey The University of Arkansas for Medical Sciences, United States
REVIEW RETURNED	27-Jun-2019

GENERAL COMMENTS	This is nice work that will add to the field. There were just a few things that could be clarified in the study protocol. 1. Why were the ages of the children restricted to 4-18? The ASEBA/CBCL can be used as early as 18 months. Greater rationale for the ages of the children included would be helpful. At the high end of the age range, there seems to be some potential for confounding. If exposure for children is measured at 18, educational outcomes would be retrospective (i.e., outcomes would be measured before exposure)?2. In the discussion of moderators, the ones used by Riglin et al are relatively unmalleable (age, gender, length of follow up) while
---

	the ones proposed in the current protocol are very malleable (executive function, sleep, classroom environment and engagement, family environment and parent involvement). How will the analyses account for what could be vast heterogeneity in when the moderator is measured? Are they expected to be measured at the follow up when educational outcomes are attained (meaning they would be included more as controls/covariates)? 3. It would seem that controlling/covarying concurrent depression measured at the time of the educational outcomes would be necessary to discuss causal relationships, but there is no mention of that in the protocol. 4. RCTs are excluded from the study, but use of data from the comparison group might enlarge the sample of studies included.
--	---

REVIEWER	Candyce Hamel Ottawa Hospital Research Institute, Canada
REVIEW RETURNED	08-Aug-2019

GENERAL COMMENTS	Please find my feedback below. Generally, I think there are several limitations in the description of the methods, as well as in how the review itself will be performed. One main point is that if the review was set to be completed in July 2019, I'm not sure there is a point in publishing this protocol. Any of the suggestions I've made will be moot as they will have already been done, as it is now August 2019.  - The introduction is more of an objective than an introduction and doesn't provide much background information. This could be expanded on further. - The authors states in the article summary that "limiting the review to peer-reviewed studies... will ensure that included studies are of a reasonable quality, and reduce heterogeneity for meta-analysis." This is not at all true. There are plenty of poor reported studies that are published. Additionally, saying it will reduce heterogeneity is completely inaccurate. Peer-review publication does not change the methods and criteria the primary study authors use. - The eligibility criteria should be broken down into the PICOTS elements (or a framework such as this, for example SPIDER), with inclusion and exclusion for both. I think this will help clarify some of the following questions:  o Is depression or depressive symptoms defined by validated methods and tools only? o What settings are included? You only have post-secondary as an exclusion criterion. o How are you determining which countries have compulsory education policies? Is there a list you are working from? o You state that you will include those with "full-text available". What does this mean exactly? Only those that are open access? Do you have access to a library with subscriptions to many journals? o Why wouldn't you consider retrospective studies and case-control studies? o You state that editorials, opinion pieces, letters to the editor and commentaries is excluded because they wouldn't be published in a peer-reviewed journal, but this isn't the case.
---

	 - How often will you contact the corresponding authors? How will you contact them? - Information sources: What will you do if experts and corresponding authors send unpublished work? Will you ask them specifically for work already published in a peer-reviewed journal? - Search strategy for electronic databases: Who developed the search strategy? Was it a library scientist? Will you get the search strategy PRESSed? (not mandatory, but good practice) - Data management: I would suggest not using Microsoft Excel for screening. This software is not meant for that purpose and does not provide a transparent approach. There is free software available meant for screening (e.g., Abstrackr). - Selection process: I would not use the 10% rule for screening piloting. If you have a large number of records at title and abstract (e.g., 10000) you would have to screen a lot prior to checking for agreement. Using a number, for example 100 at title and abstract and 25 at full-text is much more feasible. - Data extraction  o Will you do a pilot for data extraction? o Since you are including observational studies, will you also record any variables used for adjusting for confounding? o What will you do with studies that use the same data source, but just report results at different time periods? This will be double counting if you include both. - Risk of bias: Will you do a pilot for this? - Again, in the discussion section, peer-review does not guarantee quality. This fact has been studied and published.
--	---

VERSION 1 – AUTHOR RESPONSE

Reviewer: 1

Reviewer Name: Mary N Haan

Institution and Country: UCSF School of Medicine, San Francisco, CA, USA

Please state any competing interests or state 'None declared': None declared

This is a plan for a systematic review of the literature about the association between childhood depression and later educational attainment. The review is expected to be completed by July 2019. Selection criteria for studies are appropriate and result in the inclusion of only longitudinal studies.

Thank you for your positive endorsement of our review design.

Data on parents is not included in the description; this could be essential in that the childhood environment is known to powerfully affect subsequent mental health and other outcomes. The parental social context, parental history of depression/depressive symptoms, early parental death/abandonment/rejection, and childhood socioeconomic status are all of critical importance.

These are not mentioned in the review.

We agree that this could be critical, and if included studies report on these factors they will be captured as part of the information we collect on covariates, moderators and mediators (Supplement 2). We have added further detail on this as you suggest (pages 3 and 7).

The influence of educational attainment is intended to be an outcome in this study; however, education commences at age 4-5 and mental health context should be modeled as a time dependent variable.

We are interested in attainment subsequent to depression, so have limited scope to explore the more nuanced timing aspects of education and mental health. We have added this to our limitations section, but agree that this would be interesting for future work (page 8).

Reviewer: 2

Reviewer Name: Lorraine McKelvey

Institution and Country: The University of Arkansas for Medical Sciences, United States

Please state any competing interests or state 'None declared': None declared

This is nice work that will add to the field. There were just a few things that could be clarified in the study protocol.

Thank you for this endorsement of our review, we are pleased the reviewer has found that our proposal will contribute to the field.

1. Why were the ages of the children restricted to 4-18? The ASEBA/CBCL can be used as early as 18 months. Greater rationale for the ages of the children included would be helpful. At the high end of the age range, there seems to be some potential for confounding. If exposure for children is measured at 18, educational outcomes would be retrospective (i.e., outcomes would be measured before exposure)?

The age restriction applies only to exposure measurement. Because we will be looking at the effect of depression on educational attainment, we are primarily interested in depression which occurs during the school years. We have added this clarification on page 4.

The "quantitative longitudinal studies with prospective data collection" inclusion criteria must be met; therefore if a study measures depression at age 18 as you speculate, the educational outcome of interest would need to relate to qualifications received after age 18, such as completion of higher education or similar. We've added clarification that educational attainment as measured during adulthood would be permissible (page 4).

2. In the discussion of moderators, the ones used by Riglin et al are relatively unmalleable (age, gender, length of follow up) while the ones proposed in the current protocol are very malleable (executive function, sleep, classroom environment and engagement, family environment and parent involvement). How will the analyses account for what could be vast heterogeneity in when the moderator is measured? Are they expected to be measured at the follow up when educational outcomes are attained (meaning they would be included more as controls/covariates)?

As with the main meta-analysis, the pooling of moderators will only be possible where similar moderator variables or analyses are available for multiple studies (for instance, if participant characteristics are measured on similar scales and adjusted for at similar timepoints). These investigations will be informed post hoc by the included studies. We have added this clarification on page 7.

Even if they cannot be combined in meta-analysis, information on moderators and covariates will be extracted and presented (Supplement 2).

3. It would seem that controlling/covarying concurrent depression measured at the time of the educational outcomes would be necessary to discuss causal relationships, but there is no mention of that in the protocol.

We agree that adjustment for covariates will strengthen causal inference. We have added this and a relevant reference on page 5.

4. RCTs are excluded from the study, but use of data from the comparison group might enlarge the sample of studies included.

Yes, although even the control groups of RCTs often receive interventions which we would wish to avoid. Observational study designs, particularly longitudinal cohort studies, are the most suited for investigating these kinds of relationships (<https://www.ncbi.nlm.nih.gov/pmc/articles/PMC2924977/>), and the pilot search suggests that a good number will likely be eligible for the review irrespective of this restriction on study design. Nevertheless, we agree that we should discuss this further and have added the restriction on study design to our limitations section (page 8).

Reviewer: 3

Reviewer Name: Candyce Hamel

Institution and Country: Ottawa Hospital Research Institute, Canada

Please state any competing interests or state 'None declared': None declared

I've attached my comments in the attachment.

Please find my feedback below. Generally, I think there are several limitations in the description of the methods, as well as in how the review itself will be performed. One main point is that if the review was set to be completed in July 2019, I'm not sure there is a point in publishing this protocol. Any of the suggestions I've made will be moot as they will have already been done, as it is now August 2019. When originally submitted the review was scheduled for completion in July 2019, but we adjusted our timeline to December 2019 following discussion with the assistant editor at BMJ Open to allow time for peer review. This change was recorded in PROSPERO at the time (www.crd.york.ac.uk/PROSPERO/display_record.asp?ID=CRD42019123068), and now that we have been given opportunity to revise the manuscript we have amended the date accordingly.

We are very grateful your comments – they have been helpful for adding clarity to the protocol.

- The introduction is more of an objective than an introduction and doesn't provide much background information. This could be expanded on further.

We have added more background information to our introduction accordingly (page 3).

- The authors states in the article summary that "limiting the review to peer-reviewed studies... will ensure that included studies are of a reasonable quality, and reduce heterogeneity for metaanalysis."

This is not at all true. There are plenty of poor reported studies that are published.

Additionally, saying it will reduce heterogeneity is completely inaccurate. Peer-review publication does not change the methods and criteria the primary study authors use.

Thank you for pointing this out, these comments were intended to be in reference to the use of standardised depression measures and academic records which we draw attention to in that same sentence. We have removed reference to peer-review from this bullet point for clarity.

- The eligibility criteria should be broken down into the PICOTS elements (or a framework such as this, for example SPIDER), with inclusion and exclusion for both. I think this will help clarify some of the following questions:

Thank you for this suggestion, we have structured our eligibility criteria according to PECOS on page 4 accordingly.

o Is depression or depressive symptoms defined by validated methods and tools only?

Yes, as stated on page 5: "Child or adolescent depression will be operationalised as depressive symptoms or depression diagnosis as identified using a standardised diagnostic measure or a named measurement instrument."

o What settings are included? You only have post-secondary as an exclusion criterion.

There will be no restrictions imposed on study setting other than including countries with compulsory educational policies and excluding post-secondary education settings. We have added this to the text (page 4).

o How are you determining which countries have compulsory education policies? Is there a list you are working from?

This will be verified from the countries' government or public sector websites. We have added this to the text (page 4).

o You state that you will include those with "full-text available". What does this mean exactly? Only those that are open access? Do you have access to a library with subscriptions to many journals?

To be included, the full text needs to be available either publicly, through King's College London's institutional access (subscriptions to over 25,000 electronic journals), or following full text requests sent to study authors. We have elaborated this in the text (page 5).

o Why wouldn't you consider retrospective studies and case-control studies?

This is to aid the inference of a causal direction. Retrospective studies and case-control studies are at increased risk of recall bias. We have added a citation for clarity (page 5). Nevertheless, we agree that we should discuss this further and have added the restriction on study design to our limitations section (page 8).

o You state that editorials, opinion pieces, letters to the editor and commentaries is excluded because they wouldn't be published in a peer-reviewed journal, but this isn't the case.

These types of publication will be excluded because they do not represent original research, as stated earlier in the same sentence. We have parsed this sentence for greater clarity (page 5).

- How often will you contact the corresponding authors? How will you contact them?

The corresponding authors will be contacted twice, and via email (added on page 5).

- Information sources: What will you do if experts and corresponding authors send unpublished work? Will you ask them specifically for work already published in a peer-reviewed journal?

In emails, we will include a link to the PROSPERO record, which details the inclusion criteria (added on page 5). Any unpublished work received irrespective of this will not be eligible for inclusion in the study.

- Search strategy for electronic databases: Who developed the search strategy? Was it a library scientist? Will you get the search strategy PRESSed? (not mandatory, but good practice)

The search strategy was developed by the lead author. We have added this to 'Author contributions' on page 9. Co-authors and collaborators named in acknowledgements provided comments on the study design, including search strategy. King's College London's library services were also consulted on some queries which arose when designing the search strategy. The authors have prior experience of developing search strategies for published systematic reviews.

Our timeline does not allow for additional peer review beyond comments received on this protocol, but we will bear PRESS in mind for future reviews.

- Data management: I would suggest not using Microsoft Excel for screening. This software is not meant for that purpose and does not provide a transparent approach. There is free software available meant for screening (e.g., Abstrackr).

Templates have been developed which ensure transparent systematic review screening in Microsoft Excel (e.g. <https://guides.lib.unc.edu/systematic-reviews/tools-resources>), and the BMJ has previously published reviews which used Excel to track screening (e.g. <https://www.bmj.com/content/342/bmj.d1199.full.pdf+html>). This software will be used for the ease of the reviewers who both had ready access to Microsoft Excel, and to ensure that as many people as possible can access our review screening process on request and without downloading specialist software.

- Selection process: I would not use the 10% rule for screening piloting. If you have a large number of records at title and abstract (e.g., 10000) you would have to screen a lot prior to checking for agreement. Using a number, for example 100 at title and abstract and 25 at full-text is much more feasible.

Thank you for this suggestion. However we wish to be as rigorous as possible in our screening, and initial pilot search results suggest that 10% will be feasible.

- Data extraction

o Will you do a pilot for data extraction?

Yes, as stated on page 6: "Data will be extracted using a data extraction form which will be informed by the full-text screening and will be piloted on the included studies before being finalised".

o Since you are including observational studies, will you also record any variables used for adjusting for confounding?

Thank you for this suggestion, we have added this to Supplement 2.

o What will you do with studies that use the same data source, but just report results at different time periods? This will be double counting if you include both.

As stated on page 7: "If multiple articles measure the same association in an identical cohort, [...] the mean of these associations will be taken and used in meta-analysis."

- Risk of bias: Will you do a pilot for this?

Yes, we will pilot the risk of bias assessment on 10% of the included studies before proceeding to the remaining studies (added on page 7).

- Again, in the discussion section, peer-review does not guarantee quality. This fact has been studied

and published.

Thank you for pointing this out, we have removed this justification (page 8).

VERSION 2 – REVIEW

REVIEWER	Candyce Hamel Ottawa Hospital Research Institute, Canada
REVIEW RETURNED	25-Sep-2019

GENERAL COMMENTS	Thank you for your edits and comments, the methods are much clearer. Just two suggestions: 1. Under quality assessment, you might want to state which are the first and second most important factors that you will be considering under the 'Comparability' question in the Newcastle Ottawa Scale. 2. As the search was run in November 2018, if feasible, it might be worth rerunning the search to capture any new studies since then. If it is set to be completed in December 2019, and publication may take 3-6 months, the search could be out of date by that time.
--

VERSION 2 – AUTHOR RESPONSE

Reviewer 3:

Thank you for your edits and comments, the methods are much clearer. Just two suggestions:

1. Under quality assessment, you might want to state which are the first and second most important factors that you will be considering under the 'Comparability' question in the Newcastle Ottawa Scale.

Thank you for this suggestion, we have added this on page 7.

2. As the search was run in November 2018, if feasible, it might be worth rerunning the search to capture any new studies since then. If it is set to be completed in December 2019, and publication may take 3-6 months, the search could be out of date by that time.

Thank you, we agree with this suggestion and have added our intention to update the search to the section on study status, page 8.